

# Satellite remote sensing reveals a positive impact of living oyster reefs on microalgal biofilm development

Caroline Echappé[1,2], Pierre Gernez[1], Vona Méléder[1], Bruno Jesus[1,3], Bruno Cognie[1], Priscilla Decottignies[1], Koen Sabbe[2], Laurent Barillé[1]

[1]University of Nantes, Lab. Mer Molécules Santé - EA 2160, 2 rue de la Houssinière, 44322 Nantes Cedex 3, France
[2]Ghent University, Department of Biology, Lab. Protistology and Aquatic Ecology, Krijgslaan 281 – S8, 9000 Ghent, Belgium
[3]University of Lisboa, Faculty of Sciences, BioISI – Biosystems & Integrative Sciences Institute, Campo Grande, 1749-016 Lisboa, Portugal

*Correspondence to*: Caroline Echappé (caroline.echappe@univ-nantes.fr; caroline.echappe@gmail.com)

**Abstract.** Satellite remote sensing (RS) is routinely used for the large-scale monitoring of microphytobenthos (MPB) biomass in intertidal mudflats, and has greatly improved our knowledge of MPB spatio-temporal variability and its potential drivers. Processes operating at smaller scales however, such as the impact of benthic macrofauna on MPB development, to date remain underinvestigated. In this study, we analysed the influence of wild *Crassostrea gigas* oyster reefs on MPB biofilm development using multispectral RS. A 30-year time series (1985-2015) combining high resolution (30 m) Landsat and SPOT data was built in order to explore the relationship between *C. gigas* reefs and MPB spatial distribution and seasonal dynamics, using the normalized difference vegetation index (NDVI). Emphasis was placed on the analysis of a before after control impact (BACI) experiment designed to assess the effect of oyster killing on the surrounding MPB biofilms. Our RS data reveal that the presence of oyster reefs positively affects MPB biofilm development. Analysis of the historical time series first showed the presence of persistent, highly concentrated MPB patches around oyster reefs. This observation was then confirmed by the BACI experiment which showed that killing the oysters (while leaving the physical reef structure, i.e. oyster shells, intact) negatively affected both MPB biofilm biomass and spatial stability around the reef. As such, our results are consistent with the hypothesis of nutrient input as an explanation for the MPB growth promoting effect of oysters, whereby organic and inorganic matter released through oyster excretion and biodeposition stimulates MPB biomass accumulation. MPB also showed marked seasonal variations in biomass and patch shape, size and degree of aggregation around the oyster reefs. Seasonal variations in biomass, with higher NDVI during spring and fall, were consistent with those observed at broader scales in other European mudflats. Our study provides the first multi-sensor RS satellite evidence of the promoting and structuring effect of oyster reefs on MPB biofilms.

# 1 Introduction

The Pacific oyster *Crassostrea gigas* (Thunberg, 1793) is one of the most cosmopolitan marine macroinvertebrates, mainly as a result of its introduction in many countries for aquaculture purposes (Ruesink et al., 2005). In Europe, it was massively





imported in the 1970s and rapidly became the main cultivated species, following the decline of previously farmed oysters which had been struck by large-scale epizootic outbreaks (Grizel and Héral, 1991; Humphreys et al., 2014). During the last decades, *C. gigas* benefited from coastal eutrophication and rising sea temperature (Thomas et al., 2016), resulting in a poleward expansion of its distribution (Dutertre et al., 2010) and the formation of dense reefs along many coastal areas

(Diederich, 2006; Brandt et al., 2008; Le Bris et al., 2016). In some ecosystems, wild *C. gigas* is now considered as a trophic competitor of its cultivated conspecifics (Cognie et al., 2006). Newly established oyster populations also impact biogeochemical fluxes and ecosystem processes, leading to both positive (e.g. nutrient recycling) and negative (e.g. biomass regulation) impacts on local primary producers (Prins et al., 1997; Troost, 2010).

In intertidal mudflats, the main primary producers are benthic microalgal assemblages commonly referred to as

microphytobenthos (MPB) (MacIntyre et al., 1996). MPB develops within the upper millimetres of the sediment and migrates toward the sediment surface at low tide, forming transient photosynthetic biofilms (Kromkamp et al., 1998; Consalvey et al., 2004; Jesus et al., 2009; Perkins et al., 2010; Coelho et al., 2011). MPB can contribute up to 50% of total primary production in estuarine and coastal ecosystems (Underwood and Kromkamp, 1999) and is an important food source for higher trophic levels (Miller et al., 1996). After resuspension in the water column (De Jonge and Van Beuselom, 1992), benthic microalgae

become available for filter feeders such as oysters (Decottignies et al., 2007). However, MPB and oyster interactions are more complex than a simple predator-prey relationship. In (Colden et al., 2016), experimental oyster reefs consisting of empty shells only were shown to modify the local hydrodynamic conditions and to promote the trapping of fine particles, providing conditions more conducive to benthic microalgal development. (Dame and Libes, 1993) and (Newell, 2004) suggested that oysters stimulate microalgae growth through nutrient inputs derived from the mineralization of oyster excretion products (feces

and pseudofeces). It is as yet unclear however to what degree the growth promoting effects are caused by the physical structure of the reef and/or the biological activity of live oysters.

At low tide, MPB biomass can be quantified with the Normalized Difference Vegetation Index (NDVI) (Méléder et al., 2003; van der Wal et al., 2010; Brito et al., 2013; Benyoucef et al., 2014) which uses chlorophyll *a* optical properties in the red and near-infrared spectral regions. MPB biomass is characterized by high heterogeneity occurring at various spatial scales

(Chapman et al., 2010). Intra- and interannual MPB variability have previously been assessed at different scales using archived satellite images. For example, MPB seasonal cycles were described for several European mudflats using either medium resolution (250 m) satellite data such as MODIS (van der Wal et al., 2010), or higher spatial resolution (10 - 20 m) data such as SPOT (Brito et al., 2013). Few RS studies to date however have described MPB dynamics at both high spatial and temporal resolution, due to the reduced availability of high spatial resolution satellite time series together with the constraints related to

the acquisition of RS data during low tide only. As a result, factors driving MPB spatio-temporal patterns at the mudflat scale are not yet fully understood. To our knowledge, MPB spatial structure and temporal variability have never been specifically addressed in relation to shellfish populations, although preliminary analysis of SPOT images suggested that oyster-farming proximity might positively affect MPB concentration (Méléder et al., 2003).



In the present study, a unique dataset of Landsat and SPOT images acquired at low tide were combined into a 30-year time series in order to study MPB spatio-temporal variations in relation to wild oyster reefs. This high resolution satellite time series was used to 1) characterize the spatial distribution of MPB biomass around intertidal oyster reefs, and 2) investigate the impact of oysters on MPB biomass dynamics during an *in situ* ecological field experiment. The experiment consisted in the killing of wild *C. gigas* oysters from a reef surrounded by a clearly identified MPB patch. The preservation of the physical structure of the reef itself allowed to specifically focus on the influence of live oysters on MPB biomass development by the means of a before-after control-impact analysis.

## 2 Material & Methods

### 2.1 Study site and experiment

Bourgneuf Bay is a macrotidal bay located south of the Loire estuary on the French Atlantic coast (47°02'N, 2°07'W) (Fig. 1), containing large intertidal mudflats (100 km²) colonized by microphytobenthic biofilms. The site is characterized by the extensive aquaculture of the Pacific oyster *Crassostrea gigas* (Thunberg, 1793). Oyster farms cover about 10% of the intertidal area, while most of the rocky areas (about 17% of the intertidal area) are colonized by wild oysters (Le Bris et al., 2016) or macroalgae (Combe et al., 2005).

MPB spatio-temporal distribution and spatial associations with oysters were analysed using RS of biofilms developing around wild oyster reefs. Perennial MPB biofilms dominated by epipelic diatoms (Barillé et al., 2007) were previously observed in the same area using satellite data (Méléder et al., 2003). In addition, in order to investigate the effect of oysters on MPB spatial distribution, an experiment was conducted following a Before After Control Impact (BACI) design (Stewart-Oaten et al., 1986) on two *C. gigas* wild oyster reefs surrounded by MPB biofilms (Fig. 2). The reefs were selected to meet the following requirements: comparable size and oyster biomass, subject to similar environmental conditions and located at the same bathymetric level (between +2 m and +3 m above chart datum), in an area distant from oyster farms. Based on GPS field measurements and photo-interpretation, the surface colonized by oysters was estimated to be 1044 m² for the first reef and 894 m² for the second reef. Before the experiment, their stocks of wild oysters were estimated at around 23 and 20 tons respectively (Le Bris et al., 2016).

The BACI experiment was set up as follows. The first reef (hereafter designated by "R1"; left reef on Fig. 2a) was used as a control, while the oysters colonizing the second reef (hereafter designated by "R2"; right reef on Fig. 2a) were killed. This was achieved by bringing straw by boat and covering the reef with it, then setting fire to it during low tide, over two consecutive days on 16th and 17th July 2014. The burning of the reef allowed to kill the oysters, while the physical structure of the reef itself and its shells remained intact. The impact of the oyster killing on the spatial distribution of MPB biomass around R1 and R2 before and after the experiment was assessed using satellite imagery. In addition to the analysis of spatial and seasonal patterns where the experiment took place (0.36 km²), hereafter referred to as "the experimental site" (Fig. 1, 2a), a larger area was also



analysed to assess MPB seasonal variability at the scale of the mudflat (42.65 km²), hereafter referred to as "the mudflat area" (dashed area in Fig. 1).

## 2.2 Satellite data

A thirty year time series (1985 - 2015) of satellite data was built using a combination of Landsat and SPOT data (Table 1). Landsat 5 and 8 data were downloaded from the US Geological Survey (USGS) Earth Explorer data portal

(http://earthexplorer.usgs.gov/). Most SPOT data were acquired on demand by the CNES (Centre National d'Etudes Spatiales) and Airbus Defence & Space, with the exception of the Take 5 data (Hagolle et al., 2015b) which were freely available from the Theia Land Data Centre web portal (https://www.theia-land.fr/en). The Theia data portal was also used to download some Landsat 5 data.

The originality of the dataset lies in its high spatial resolution (6 to 30 m), its long time duration (30 years) and the high

number of selected images (47) due to the combination of several satellite missions (Table 1). All Landsat and SPOT sensors display only slight variations in the position and width of the red and the near infrared (NIR) bands (Table 1), allowing for the calculation of comparable NDVI values (formula in section 2.3) as a chlorophyll *a* proxy. NDVI interconsistency between Landsat and SPOT sensors was estimated using a synthetic hyperspectral library of benthic diatoms (Barillé et al., 2011). Library reflectance spectra covering a wide range of diatom biomass over different types of sediment were downscaled to each

sensor spectral resolution and compared.

Careful data quality-control was performed on the initial dataset. Data were first selected according to their acquisition parameters: images with a cloud cover higher than 10% above the experimental site and the mudflat area were eliminated. Considering the location of the oyster reefs, images with a water height (based on LAT) superior to 2.5 m at the nearest reference harbour (Pornic harbour) at the time of the acquisition were excluded. In daylight, reflectance-based estimates of

MPB biomass at the sediment surface can vary during tidal emersion due to vertical migration of the microalgae in the surface sediment layers (Serôdio et al., 1997). These migratory rhythms result in MPB accumulating at the sediment surface around mid-low tide, and decreasing during ebb and flow. The latter will result in lower NDVI values. To avoid biases introduced by this phenomenon, the impact of image acquisition time during low tide on NDVI values was investigated. Images for which the NDVI was found to be impacted by MPB vertical migration or mudflat submersion (i.e. resulting in a low percentage of

uncovered mudflat and/or abnormally low NDVI) were identified using the following criteria: water height, timing of low tide, MPB pixel count (total number of pixels identified as MPB within an image, i.e. excluding rocks, oyster reefs, water and macroalgae, see below), and outlier NDVI values. Images which did not follow a normal distribution based on each of these criteria were excluded. At the end of the quality-control process, a total of 47 images acquired from 1985 to 2015, in a time range between 09:51 and 11:36 UT, were selected to study MPB spatial and temporal patterns at the experimental site (see

Table S1). Due to the presence of clouds at macroscale, a total of 44 images could be used for MPB analysis at the scale of the mudflat area.



### 2.3 Data processing

#### 2.3.1 Satellite data processing

Landsat 5 (2009 to 2011) and SPOT Take 5 land surface reflectance products using the multisensory atmospheric correction
and cloud screening method (MACCS, Hagolle et al., 2015a) were directly available from the Theia web portal. Other Landsat
and SPOT images were atmospherically corrected and converted into surface reflectance with the Fast Line of sight
Atmospheric Analysis of Spectral Hypercubes (FLAASH, (Matthew et al., 2000) method using ENVI 5.1. For coherence
within the time series, the same FLAASH parameters (US atmospheric model, 40 km initial visibility, maritime aerosol model)
were applied. The NDVI was then calculated from surface reflectance following Eq. (1):

$$NDVI = \frac{R_{NIR} - R_{Red}}{R_{NIR} + R_{Red}},$$  (1)

where $R_{NIR}$ and $R_{Red}$ are respectively the reflectance in the near infrared and red regions (Table 1). All data were projected in
the WGS84 UTM30N coordinate reference system and downscaled to the lowest spatial resolution (30 m, Landsat resolution)
by applying an inverse distance weighted interpolation. All statistical analyses were carried out on these downscaled data.

Multispectral RS does not allow for the differentiation between micro- and macroalgae, leading to possible confusion
between high MPB and low macroalgal biomasses (van der Wal et al., 2010)(van der Wal et al., 2014). In this study, MPB
biofilms were discriminated using two methods. First, a geometric mask was applied to the rocky areas in order to eliminate
most macroalgae and epilithic microalgae (Le Bris et al., 2016). Secondly, a radiometric mask was applied to negative NDVI
values to exclude water pixels and to NDVI > 0.4 to exclude macrophytes found on sediments. The latter threshold was chosen
according to the maximum NDVI values observed on pixels corresponding to known MPB biofilms.

Spatial data analysis was carried out with R software (R Core Team, 2015) using gstat (Pebesma, 2004), maptools (Bivand
and Lewin-Koh, 2015), raster (Hijmans, 2015), rgdal (Bivand et al., 2015), rgeos (Bivand and Rundel, 2015), and sp (Pebesma
and Bivand, 2005; Bivand et al., 2013) packages.

#### 2.3.2 Satellite data ground truthing

Spectroradiometric field measurements were performed in order to ground-truth the satellite-derived NDVI data (Forster and
Jesus, 2006). Due to the difficulty of access, the experimental site was sampled only once and a nearby site (La Coupelasse,
1.5 km further) was sampled twice for matchup purposes (Table 2). An ASD FieldSpec 3FR spectroradiometer was used to
measure the *in situ* radiance (mW cm$^{-2}$ nm$^{-1}$ sr$^{-1}$) in the 350-2500 nm spectral range. Reflectance was calculated by dividing
the surface radiance by the downwelling radiance measured with a 99% reflectance standard panel (Spectralon® plate).
Hyperspectral reflectance data were then downscaled to the resolution of the matching satellite data using the sensor spectral
response function, and NDVI was calculated. A total number of 57 ground-truth stations were obtained from *in situ* transects
conducted during the three sampling campaigns. For each matchup station, three replicates per 30 x 30 m satellite pixel were
measured *in situ*, averaged and compared to the corresponding satellite NDVI pixel.



### 2.3.3 Spatial analysis of MPB around oyster reefs

At the scale of the experimental site, NDVI spatial distribution around the two oyster reefs was characterized using transect

analysis (dashed line through R1 and R2 on Fig. 2) and spatial statistics. Well-defined MPB spatial structures were recurrently observed around the reefs, hereafter referred to as NDVI patches. These patches were delimited following the boundary detection method (Dale and Fortin, 2014), by defining a NDVI minimum threshold value allowing to connect pixels of a common value in a closed contour line. This delimitation was performed using an algorithm applied independently on each image and reef. Patch spatial properties (area, shape, patchiness) were then extracted independently for each image using

spatial metrics calculated with the R SDMtools package (VanDerWal et al., 2014) based on the patch statistics provided by the FRAGSTATS software (McGarigal et al., 2012). Each patch was described by measuring its area and, as respectively calculated following Eq. (2) and Eq. (3), the fractal dimension index (FDI) and aggregation index (AI) (McGarigal et al., 2012):

$$FDI = \frac{2\ln(.25\,p_{ij})}{\ln a_{ij}}, \tag{2}$$

$$AI = \left[\frac{g_{ij}}{max{\rightarrow}g_{ij}}\right](100), \tag{3}$$

with $p_{ij}$ the perimeter (m) of patch ij, $a_{ij}$ the area (m²) of patch ij, $g_{ij}$ the number of like adjacencies (joins) between pixels of patch type i based on the single-count method, and max→ $g_{ij}$ the maximum number of like adjacencies (joins) between pixels of patch type i based on the single-count method. The FDI allows to characterize patch shape complexity, with a value of 1 indicating very simple perimeters, and a value of 2 representing highly convoluted perimeters. The AI, expressed as a

percentage, defines the percentage of patch spatial aggregation, with 0 % expressing a maximally disaggregated patch, and 100 % a patch maximally aggregated into a single, compact patch.

In order to extract average NDVI values associated with both patches (around R1 and R2) throughout the time series, a "distance buffer" was created for each reef. From the patch areas determined in each image as described above, average areas were computed for each patch. These average areas were then used to apply isotropic, fixed-distance buffers of the same

average area around each reef across the whole time series. NDVI values included within the distance buffers were extracted and averaged for each reef throughout the time series. MPB biomass response to the oyster killing was analysed by comparing R1 and R2 NDVI average values before and after the experiment. This method allowed to exclude the background noise induced by natural spatial and temporal variability in MPB (e.g. related to seasonal development), by focusing on the biomass variation recorded between the control (R1) and the impacted reef (R2).

A composite monthly signal over the 1985-2015 time series was determined by clustering and averaging NDVI monthly data across the mudflat area. No data were available for December and January due to image acquisition technical constraints during the winter period (e.g. sun elevation).



### 2.4 Statistical analysis

All data processing, statistical analyses and graphical results were performed using R software (R Core Team, 2015). NDVI
normality was tested using the car package (Fox and Weisberg, 2011) and the Shapiro-Wilk normality test (p = 0.34, n = 44
for the mudflat area; p = 0.64, n = 47 for the experimental site). Correlation between *in situ* and satellite NDVI was tested
using Pearson product-moment correlation and the slope of the linear regression model applied to the data was compared to
the isometric relation x=y. The Root Mean Squared Error (RMSE) of predicted values (satellite data) vs. observed values (*in
situ* data) was calculated using the Metrics package (Hamner, 2012) and also used to assess multi-sensor interconsistency.
Mean NDVI differences were tested with Student's t-test for two samples and Kruskal-Wallis Rank Sum Test for multiple,
unbalanced samples.

### 3 Results

### 3.1 Multi-sensor RS of intertidal mudflats

### 3.1.1 SPOT and Landsat interconsistency

No significant differences were found in NDVI values from the diatom library downscaled to the satellites' spectral resolution
(Kruskal-Wallis Rank Sum Test, p = 0.62, n = 93). The coefficient of determination of all sensors' regressions was very high
($R^2$ > 0.99, p < 0.001, n = 93) with a slope not significantly different from 1 and a RMSE systematically lower than 0.03 (n =
93) (Table 3).

Sensor interconsistency was further verified by comparing SPOT and Landsat mean NDVI throughout the time series. No
significant difference was found at the scale of the mudflat area (Student's t-test, p = 0.72) nor at the scale of the experimental
site (p = 0.74). Satellite data accuracy evaluated using *in situ* ground-truthing showed a significant correlation ($R^2$ = 0.73, p <
0.001) between *in situ* and satellite NDVI (Fig. 3).

### 3.1.2 Influence of tidal stage on intertidal mudflat MPB RS

Satellite acquisition time in relation to the emersion period appeared to affect satellite NDVI measurements (Fig. 4). Unusually
low NDVI values resulting either from MPB vertical migration and/or from partial mudflat submersion were detected on
images acquired more than 100 minutes before and after low tide (black symbols in Fig. 4). These data were removed after
which the remaining NDVI data (grey symbols in Fig. 4) were not correlated any more with any of the following tidal variables:
acquisition time relative to low tide (Spearman correlation, r = 0.22, p = 0.15), water height (Pearson correlation, r = 0.20, p =
0.19), tide amplitude (Pearson correlation, r = -19, p = 0.21) and MPB coverage, as expressed by the number of pixels
considered as MPB on the images (Pearson correlation, r = -11, p = 0.49). At the time of the acquisition, the average water
height at the nearest reference harbour (Pornic, France) was 1.43 m (in a range from 0.34 to 2.29 m). For comparison, water
height can be as high as 6 m during high tide periods.

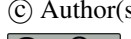



### 3.2 MPB spatio-temporal variability around oyster reefs

#### 3.2.1 MPB spatial distribution at the mudflat scale

On average MPB covered 60.8 % ± 10.4 (mean ± S.D.) of the whole mudflat area. Over the 1985-2015 time series MPB mean NDVI was 0.16 ± 0.02 for the mudflat area. Similar average values were observed for the experimental site. MPB spatial distribution generally showed regular patterns associated with bathymetry changes: NDVI maxima were consistently observed at about 2 m above chart datum, whereas NDVI minima were located on the upper and lower shore. Visual interpretation of the RS images suggested that high NDVI values coincide with the proximity of oyster farms and wild oyster reefs (see Fig. 225 S1).

#### 3.2.2 MPB spatial distribution around oyster reefs

Within the experimental site, NDVI spatial distribution was characterized by the existence of clearly defined patches around the oyster reefs (Fig. 2a). The identification of a patch around R1 and R2 was possible in respectively 97 % and 79 % of the images acquired before the BACI experiment. Patch mean NDVI was systematically higher than the average NDVI value over 230 the whole experimental site. No correlation was found between patch average NDVI and patch area neither for R1 ($R^2$ = 0.03, p = 0.22) nor R2 ($R^2$ = 0.02, p = 0.46). Distance buffers with diameters of 192 m and 128 m were determined for R1 and R2 respectively (see 2.3.3).

The average NDVI R1-R2 transect (dashed line in Fig. 2a) extracted from the data acquired before the BACI experiment highlights the influence of the oyster reefs on MPB spatial distribution (Fig. 5), with NDVI being consistently higher closer to 235 the reefs, and decreasing with distance from each reef. A NDVI minimum was clearly observable between the two reefs, at about 150 m from R1 and 60 m from R2. Two transects crossing the reefs perpendicular to the bathymetric lines (see Fig. S2) also showed high NDVI over a distance up to 150 m on either side of the reefs, with higher values recurrently observed for lower bathymetry, i.e. inferior to 2.5 m.

MPB FDI varied from 1 to 1.2, indicating that patches were characterized by simple and regular shapes. MPB AI ranged 240 from 67 to 100%, showing that the patches were generally very compact. Their shape was circular to elliptic most of the time, and in the latter case always expanded perpendicular to the bathymetric lines.

#### 3.2.3 Seasonal variability

The NDVI showed significant seasonal variations at the scale of the mudflat area, the experimental site, around the control reef R1 and the impacted reef R2 (Kruskal-Wallis rank sum test, p < 0.05, p < 0.05, p < 0.01 and p < 0.05 respectively) (see 245 Fig. S3). MPB patch around R1 exhibited two pronounced MPB biomass peaks in April and October (Fig. 6a).

The spatial metrics of the MPB patches around R1 (Fig. 6b, c and d) and R2 (not shown) also showed clear seasonal variations, together with high interannual variability as suggested by the pronounced error bars during the spring and autumnal periods. Patch areas around R1 and R2 were larger in spring than in summer, and the largest patches were observed in autumn,



from September to November (Fig. 6b). Variations in patch area were associated with variations in patch structure, with high

correlations between area and FDI (r = -0.6, p = 0.07), and between the area and the AI (r = 0.74, p < 0.05) (Fig. 6c, d). In general, larger patches corresponded to simpler (i.e. low FDI) and more aggregated shapes. Patches were especially compact around the reefs in September and October. From May to August, when MPB patch areas were smallest, their shapes became more complex (higher FDI) and less aggregated. Similar seasonal variations were generally observed for MPB patches in R1 and R2, but sometimes no patch could be delineated around R2 from March to June, corresponding to the spring – early

summer period, despite high NDVI values. This was caused by the fact that the MPB biofilm around the reef was highly disaggregated (see also below).

### 3.3 Influence of oyster reefs on MPB: a BACI analysis

### 3.3.1 NDVI variations

During the autumn following the oyster killing, high mean NDVI values were observed around R1. In contrast, values around

R2 tended to decrease, and in October 2014 the mean NDVI around R2 was below the pre-experiment monthly average value (Fig. 7). NDVI then gradually recovered to its usual level in the following months and even reached values higher than the pre-experiment average about one year later, from July to September 2015, despite the absence of a clearly identifiable patch structure around R2. This high NDVI was also observed around R1.

In order to discriminate the impact of the experiment from potential inter-site variations, the difference of the averaged

NDVI within the R1 and R2 buffers (hereafter noted Δ NDVI) was analysed before and after impact. Throughout the time series before the experiment, the average difference was 0.01 ± 0.02 (mean ± SD) (Fig. 8). The difference was significantly higher on the three images acquired within three months following the oyster killing (Δ NDVI was respectively 0.11, 0.09 and 0.07 in August, October and November 2014), as well as 9 months later in April 2015 (Δ NDVI of 0.09). The difference ratio is especially striking during the first months after the killing, considering that NDVI values around R1 were exceptionally high

in 2014 compared to the seasonal average, and that one would therefore expect R2 NDVI values to also stand above average. The difference then progressively disappeared, and from May 2015 Δ NDVI was mostly within the range of values observed before the oyster killing.

### 3.3.2 Alteration of MPB spatial structure

The killing of the oysters colonizing R2 in July 2014 was followed by an alteration of MPB biomass and spatial distribution

in the short and medium term. A clear change in spatial structure was observed for the first time three months after the biomanipulation on an image acquired during October 2014, with an increase in patch shape complexity and a decrease in aggregation percentage (Fig. 7). The impact was even more pronounced one year after the experiment as no patch could be identified around R2 on five consecutive images from July to September 2015. While MPB could still be observed in the





vicinity of the reef, its structure was unusually disaggregated and did not allow for the detection of a clear patch. This had

never been observed during this specific period of the year throughout the whole time series.

## 4. Discussion

A high resolution 30-year time series of Landsat and SPOT satellite data revealed the presence of persistent, highly concentrated MPB patches around two *C. gigas* oyster reefs in Bourgneuf Bay (France). Killing of oysters in one reef in a BACI experiment confirmed the positive impact of live oysters on MPB biomass development and dynamics. Taken together,

our data thus strongly suggest that the presence of live oyster reefs promotes MPB biomass development and affects MPB spatial distribution around the reef. This is consistent with the hypothesis that MPB development is stimulated by the release of dissolved organic and inorganic matter and biodeposits excreted by the oysters. As such, oysters and MPB would be connected in a local positive feedback loop with oysters "fertilizing" their main food source (Prins et al., 1997; Kasim and Mukai, 2006). Clear MPB seasonal dynamics consistent with the ecosystem location were also shown, and could be associated

to variations in MPB patch shape. This first observation of a positive effect of live oyster reefs on MPB biofilms using RS time series can yield new insights in the knowledge of MPB dynamics and the impact of aquaculture on the environment.

### 4.1 Oyster reefs influence on MPB biofilm development

By focusing on an intertidal ecosystem dominated by long-established oyster communities, our 1985-2015 RS satellite time series revealed a close relationship between oysters and MPB biofilm development, with oyster reefs being associated with

the presence of persistent MPB patches. The delimitation of such spatially explicit structures is generally not obvious when dealing with continuous variables (Jesus et al., 2005). In our study though, the systematically higher than the average MPB biomass concentrations observed around the reefs allowed us to distinguish MPB patches from the background MPB biomass. In addition, the presence of reefs had a significant effect on shape, area and degree of aggregation of the surrounding MPB patches in our BACI experiment. The negative impact of oyster killing on the surrounding MPB was reflected by its very

limited autumnal increase in biomass compared to the typical autumnal development (see 4.2) around this reef and the control reef. Until mid-spring of the following year, MPB biomass was also lower than one would expect given the average pre-experiment values of the non-impacted biofilm. *In situ* reflectance measurements performed ten months after the experiment confirmed this difference observed between the impacted and non-impacted biofilms (not shown). Oyster reefs however not only appeared to act as promoters of MPB biomass, but also as a factor structuring their spatial distribution. After the oysters

were killed, MPB patches regularly showed a more complex and disaggregated structure instead of the regular patch shape usually observed. Variations in bathymetry, tidal dynamics, sediment type, irradiance and grazing had already been documented as factors structuring MPB spatio-temporal distribution (Brotas et al., 1995; Méléder et al., 2003; Sahan et al., 2007; Ribeiro et al., 2013). However, to date there are only a few studies which have explicitly investigated the influence of



nearby macrobenthic communities on MPB dynamics (Dame, 1993; Méléder et al., 2003, 2007; Newell, 2004; Engel et al.,
310  2017).

While nutrient availability has long been considered as not limiting in intertidal mudflats (Underwood and Kromkamp, 1999), the hypothesis of nutrient inputs coming from oysters is here advanced as a possible main factor explaining the higher biomass of surrounding MPB. Via the release of organic and inorganic matter through excretion and biodeposition (Dame, 1993; Cognie and Barillé, 1999; Newell, 2004; Buzin et al., 2015), oysters can have an impact by enriching the sediment
around them, increasing nutrient availability and hence development of MPB (Dame, 1993; Garcia-Robledo et al., 2016). Oyster reefs are also known to have indirect effects such as modifying the structure of the surrounding sediment and the ambient hydrodynamic conditions, facilitating MPB establishment (Colden et al., 2016). In a recent paper on the impact of mussel beds on MPB biomass development, Engel et al. (2017) attributed the positive effect of such beds on MPB biomass to a combination of reduced hydrodynamic stress and increased nutrient availability (and also to potential changes in the
associated invertebrate community). In contrast with the observations of Colden et al. (2016) and Engel et al. (2017), our BACI experiment, in which the oysters were killed while the physical structure of the reef itself was not modified, now allows distinguishing the pure physical (hydrodynamic) effect from the biological (nutrient enrichment) effect of the oyster reefs, suggesting that the latter process is more important in our study area. In this respect, the resilience of MPB biomass development around the impacted reef observed one year after the experiment is probably due to the recolonization of the dead
reef by young oysters, following the exceptionally high recruitment which occurred during the autumn 2014 (Pouvreau et al., 2015). When measured in autumn 2015, one year after the experiment, the oysters newly colonizing the reef had already reached an average size of $32 \pm 4$ mm (n = 30), versus an average size of $54.5 \pm 17$ mm (n = 30) for adult oysters on the control site.

While the positive impact of oyster reefs on MPB biomass is clear from both the long-term satellite and the BACI
experimental data, a temporary negative effect of burning oyster biomass and straw on MPB biomass cannot be ruled out. Toxic compounds resulting from (incomplete) combustion of biomass could negatively affect MPB growth and as such reduce biomass. However, in our opinion the long-term negative effect on MPB biomass observed in the experiment is unlikely to be caused by such compounds because of the strong dilution effect caused by the daily tidal immersion and emersion at the study site. In contrast, the elimination of the above-mentioned oyster enrichment effect by oyster killing would have been a more
lasting effect as was observed in the experiment.

The use of a RS historical time series (29 years of data before the experiment and more than one year of data after it) allowed for a simultaneous and synoptic collection of data for both the control and impact sites, and enabled the differentiation of seasonal and interannual variability from the impact of the experiment. The paired monitoring of the control and impacted reefs confirmed a low inter-site variability, while their proximity submitted them to similar environmental conditions (Stewart-
Oaten et al., 1986). Thus despite the absence of further replicates, our experimental design was found to be suitable to detect any site-specific NDVI variations from the average trend (Hewitt et al., 2001). Satellite data of the control site acquired after



summer 2014 showed trends consistent with expected MPB temporal dynamics. They also showed no response related to the experiment, which confirmed their reliability as control values.

## 4.2 MPB temporal dynamics

MPB in Bourgneuf Bay exhibited pronounced large-scale seasonal dynamics together with a generally limited interannual variability, indicating a stable spatio-temporal structure over time (Ubertini et al., 2012). The combination of all images in a composite monthly signal (see Table S2) highlighted two peaks of MPB biomass occurring during spring and autumn. These variations were similar at both the reef and the whole mudflat scale, although peaks of biomass were more pronounced in the immediate proximity of the oyster reefs (see Fig. S3). Variations in MPB patch size, shape and degree of aggregation followed
these seasonal trends, with MPB spatial structure tending to aggregate into larger and more regular patches around the oyster reefs during spring and autumn. The seasonal cycle in Bourgneuf Bay also corresponds to those observed in other European mudflats. In some NW European flats, MPB maxima have been detected slightly later in spring and in September (van der Wal et al., 2010). The summer depression has been observed in some more southerly mudflats such as Marennes-Oléron Bay (France) (Cariou-Le Gall and Blanchard, 1995), the Tagus estuary (Portugal) (Brito et al., 2013), Cadiz Bay (Spain) (Garcia-
Robledo et al., 2016), as well as in the Wadden Sea (The Netherlands) (van der Wal et al., 2010; Stief et al., 2013).

## 4.3 RS as a tool to investigate multi-scale ecological processes

MPB colonizing mudflats around oyster reefs are dominated by epipelic life forms (Méléder et al., 2007) which often exhibit marked seasonal dynamics (Haubois et al., 2005). Pronounced variation in NDVI can also be observed at the scale of the tidal cycle (Méléder et al., 2003). This is confirmed by our data, which show that at about 1 ½ to 2 hours before and after low tide,
NDVI is lower. This can mainly be attributed to respectively upward and downward vertical migration of epipelic MPB after and before submersion. Two hours after low tide corresponds to an average water height of more than 2.3 m at the scale of the mudflat area, meaning that the experimental site is almost covered by the tide. MPB migration is therefore very rapid (Herlory et al., 2004; Coelho et al., 2011), happening only shortly after the water left and before it comes back, as it was observed in Méléder et al. (2003). While RS provides large datasets of images, careful selection of images with respect to tidal stage is
hence necessary to avoid observing low NDVI values related to vertical migration. Fortunately, the high speed of MPB migration allowed to work with satellite data that could be obtained during most of the low tide period. It should also be kept in mind that while MPB biomass at the surface of the sediment can change as a result of vertical migration (Brouwer and Stal, 2001; Chennu et al., 2013), it can also be affected by other exogenous factors such as temperature and irradiance (Saburova and Polikarpov, 2003; Jesus et al., 2009; Coelho et al., 2011). A better understanding of these temporal dynamics requires high
spatial and temporal resolution RS data.

In this study, the combination of SPOT and Landsat data allowed monitoring MPB dynamics across different spatial ($10^1$-$10^4$ m) and temporal (months to decades) scales. This underlines the interest of using multispectral, multi-sensor RS as a monitoring tool of MPB dynamics (Dube, 2012). A multi-sensor approach exposes to different sources of variability coming




from sensor technical features, differences in spatial and spectral resolutions, band position and width for the computation of
indices, and choice of the atmospheric correction. However, the standardization of the data and a careful quality-control allows
the building of robust and consistent satellite time series. Moreover, as NDVI is little influenced by the sediment background
(Barillé et al., 2011), satellite RS constitutes a valuable tool to map MPB spatio-temporal dynamics over a variety of muddy
and sandy ecosystems. Although it does not allow to identify the cause of the patterns observed, it provided a sufficiently
explicit spatial tool able to describe MPB structure at mesoscale and made it possible to quantify the size of the detectable reef
footprint (Giles et al., 2009) through the NDVI in our study. Satellite RS hence facilitates the general detection of
environmental and anthropogenic disturbances at large scales (Kerr and Ostrovsky, 2003) at a high spatial and temporal
resolution (Ibrahim and Monbaliu, 2011).

Data ground-truthing however remains essential when using RS (Forster and Jesus, 2006) and should be performed
according to the RS spatial resolution (Paterson et al., 1998), which was done here by adapting the sampling plan to match the
size of the sensors' pixels. The issue of MPB microscale patchiness was limited by performing replicate measurements so as
to be as representative as possible of the surface considered. Concerning MPB monitoring, Landsat and SPOT data did not
allow the differentiation between micro and macroalgae given the position of their spectral bands. The setting of empirical
thresholds and field knowledge however allowed excluding non-MPB organisms. However, mixed signals due to the spatial
association of objects showing different spectral signatures remains possible, and no field information is available for many
mudflats worldwide, stressing the need for satellite data with higher spectral resolution than SPOT and Landsat. Moreover,
while the downscaling of finer resolution data to the lowest one (i.e. Landsat 30 m spatial resolution) enabled to reduce the
issue of spatial heterogeneity between the sensors, it may result in potential loss of information. The new generation of satellite
data will enable a better analysis in both cases, with e.g. Sentinel 2 Multi Spectral Instrument providing 10 m resolution images,
more spectral bands, and high revisiting time (5 days with both Sentinel 2A and 2B combined).

*Data availability.* All data is available upon request to the authors.

*Supplement link.*

*Author contributions.* C.E. conducted the analysis, prepared the figures and wrote the manuscript. C.E. and P.G. conducted
fieldwork. P.G. and L.B. helped conducting the analysis. P.G., V.M., B.J., B.C., P.D., K.S. and L.B. contributed to the writing
of the manuscript.

*Competing interests.* The authors declare that they have no conflict of interest.





*Acknowledgments.* The authors wish to thank the CNES for the ISIS Program regarding the use of SPOT satellite products. SPOT6-7 data were acquired by Airbus Defence and Space in the frame of the MyGIC project. We also thank the pole THEIA and the CESBIO for the provision of SPOT Take5 (CNES) and Landsat (USGS) data. The ASD FieldSpec 3FR

spectroradiometer was supplied by the Laboratoire de Planétologie et Géodynamique (LPG UMR-CNRS 6112) of the University of Nantes. We thank Laurent Godet for his help with spatial metrics. We acknowledge the CNES and Région Pays de Loire for the funding of C. Echappé PhD. The oyster experiment has been designed and implemented in the framework of the COSELMAR project funded by the Région Pays de la Loire.

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



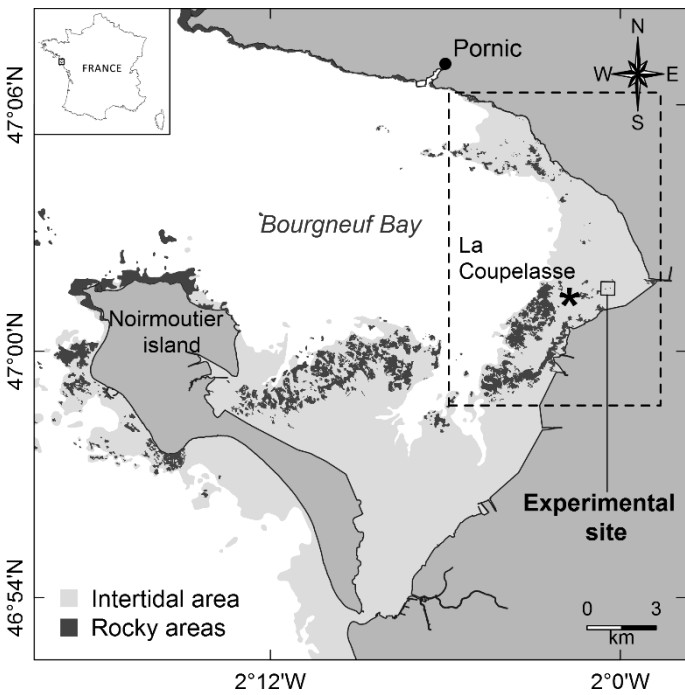

**Figure 1.** Location of Bourgneuf Bay and the experimental site. The dashed box corresponds to a mudflat with some rocky areas mainly colonized by *C. gigas* wild oysters (Le Bris et al., 2016).





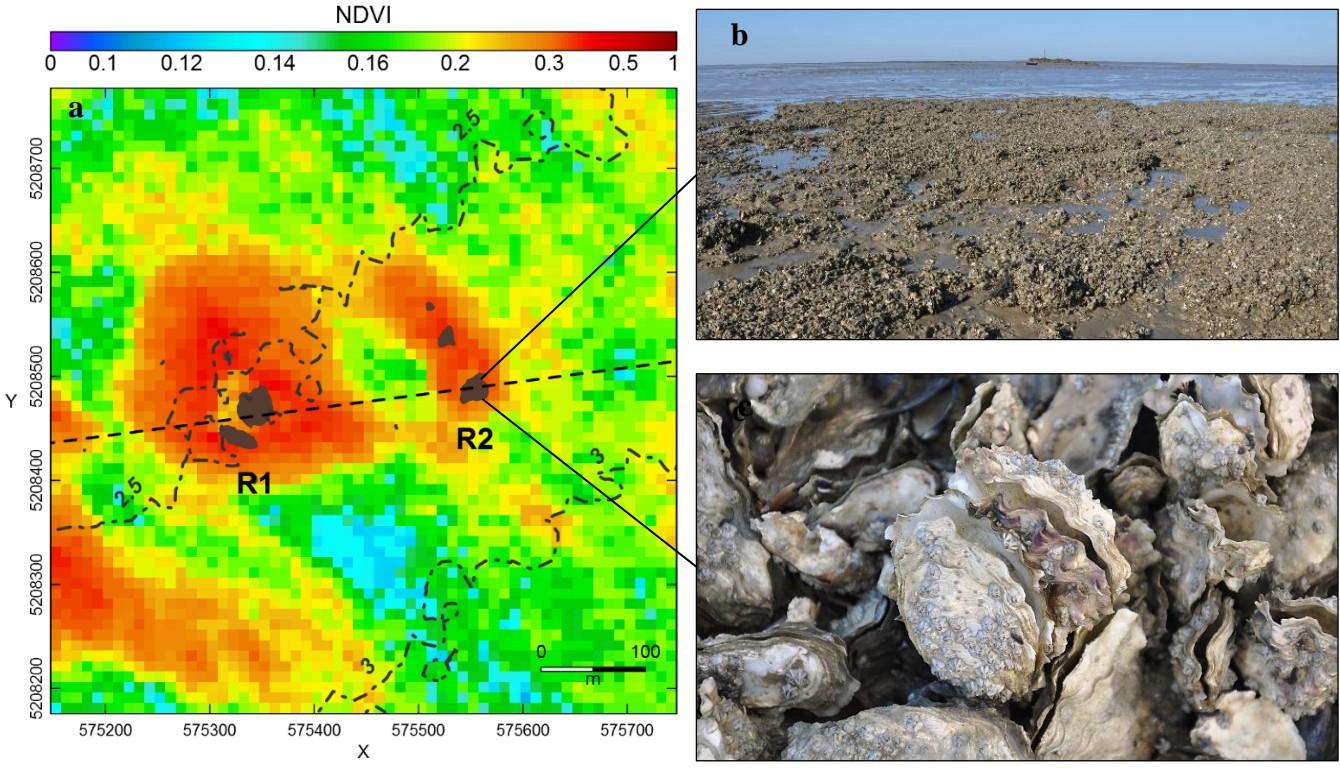


**Figure 2. (a)** NDVI map of the experimental site (SPOT 5 image acquired on 8 September 2009). The sector is 600 x 600 m, limited by a channel on the left. The control reef (R1) and the impacted reef (R2), both formed of several emerging parts, are represented in brown. The grey dotted lines correspond to the bathymetric levels. The black dashed line represents the transect used in the analysis of the biomass variations between R1 and R2. **(b)** View of the reef where the oysters were burnt (R2); R1

is visible in the background. **(c)** Open shells of dead oysters after the burning (photographs acquired in September 2014).




**Table 1.** Characteristics of the satellites missions and sensors used to build the RS time series. The number of images corresponds to the contribution of each satellite to the dataset after a quality-control data selection.

| Satellite | Mission | Resolution (m) | Red (nm) | NIR (nm) | Source | Years | Number of images |
|---|---|---|---|---|---|---|---|
| Landsat | 5 | 30 | 630 - 690 | 760 - 900 | USGS (Automatic acquisition) | 1985-2011 | 15 |
|  | 8 | 30 | 630 - 680 | 845 - 885 |  | 2013-2015 | 9 |
| SPOT | 1-4 | 20 | 610 – 680 | 780 – 890 | CNES (Acquisition on demand) | 1991-2013 | 8 |
|  | 5 | 10 | 610 – 680 | 780 – 890 |  | 2009-2015 | 5 |
|  | 6-7 | 6 | 625 – 695 | 760 – 890 |  | 2013-2015 | 10 |





**Table 2.** Field measurements performed for satellite ground-truthing.

| Site | Coordinates | Stations | Sampling date | Satellite image acquisition date | Satellite mission |
|---|---|---|---|---|---|
| Experimental site | 47°01'32"N 2°00'26"W | 28 | 18th May 2015 | 20th May 2015 | SPOT 6 |
| La Coupelasse | 47°01'14"N 2°01'44"W | 4 | 20th May 2015 | 20th May 2015 | Landsat 8 |
| La Coupelasse | | 25 | 4th June 2015 | 6th June 2015 | SPOT 6 |



**Table 3.** SPOT and Landsat NDVI interconsistency based on the reflectance spectra of a benthic diatom library.

| Compared sensors | | | $R^2$ | a | b | RMSE |
|---|---|---|---|---|---|---|
| SPOT 5 | vs. | SPOT 6 | 0.9994 (p < 0.001) | 1.01 | -0.01 | 0.00 |
| SPOT 5 | vs. | Landsat 5 | 0.9997 (p < 0.001) | 1.01 | 0.002 | 0.01 |
| SPOT 5 | vs. | Landsat 8 | 0.9988 (p < 0.001) | 1.05 | 0.005 | 0.02 |
| SPOT 6 | vs. | Landsat 5 | 0.9998 (p < 0.001) | 1.00 | 0.01 | 0.01 |
| SPOT 6 | vs. | Landsat 8 | 0.9978 (p < 0.001) | 1.03 | 0.01 | 0.03 |
| Landsat 5 | vs. | Landsat 8 | 0.9986 (p < 0.001) | 1.04 | 0.003 | 0.02 |





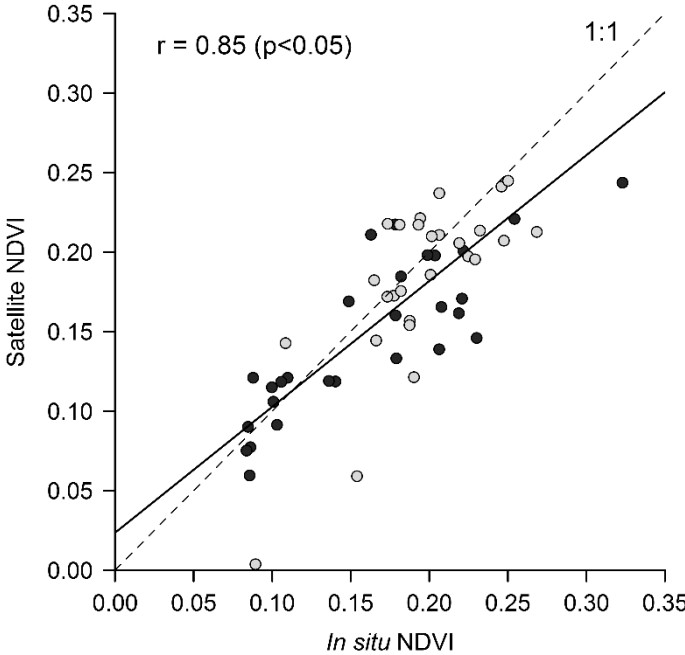

**Figure 3.** Match-up between *in situ* and satellite NDVI (see Table 2). Black dots were sampled on the experimental site, while grey dots were sampled on the additional site of La Coupelasse. Pearson correlation coefficient was calculated. The slope of the linear regression model applied to the data (black linear regression line, RMSE = 0.04, n = 57) was not significantly different from 1 (a = 0.79, p = 0.67).





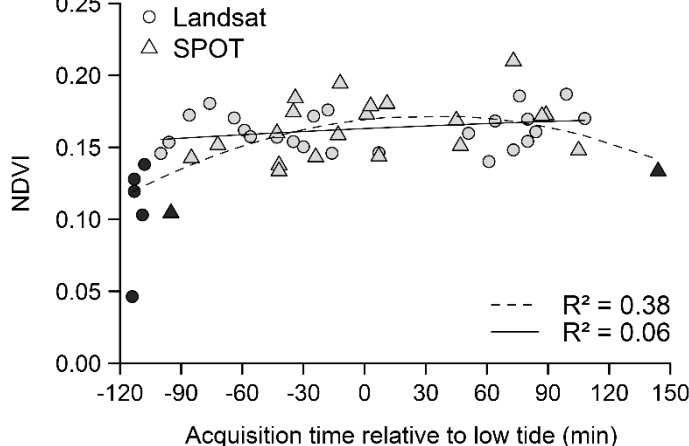

**Figure 4.** Relationship between NDVI and timing of low tide across the mudflat area. Satellite (SPOT & Landsat) data mean NDVIs are represented according to their acquisition time relative to low tide (time = 0). Black symbols correspond to data characterized by a decrease of biomass due to the process of vertical migration and/or partial mudflat submersion (dotted polynomial regression line, p < 0.001). After removing them, the regression was no longer significant (grey symbols, full polynomial regression line, p = 0.30).





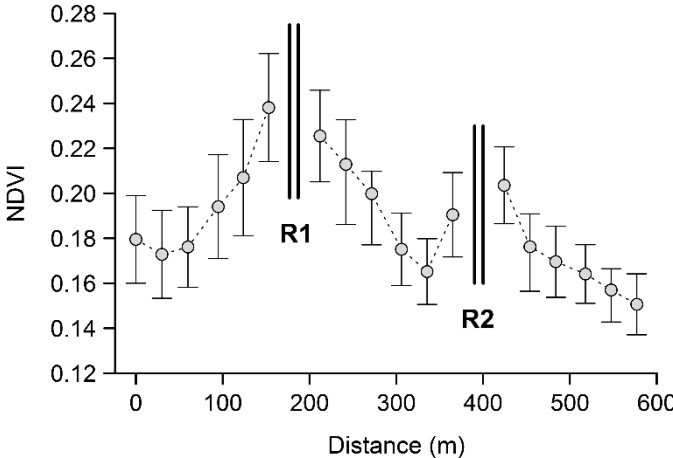

**Figure 5.** NDVI extracted from a transect going through the two oyster reefs (see Fig. 2) based on satellite data acquired before the burning (1985-2014 time series, mean ± 95% CI, n = 33). Black vertical bars correspond to the location of the reefs.





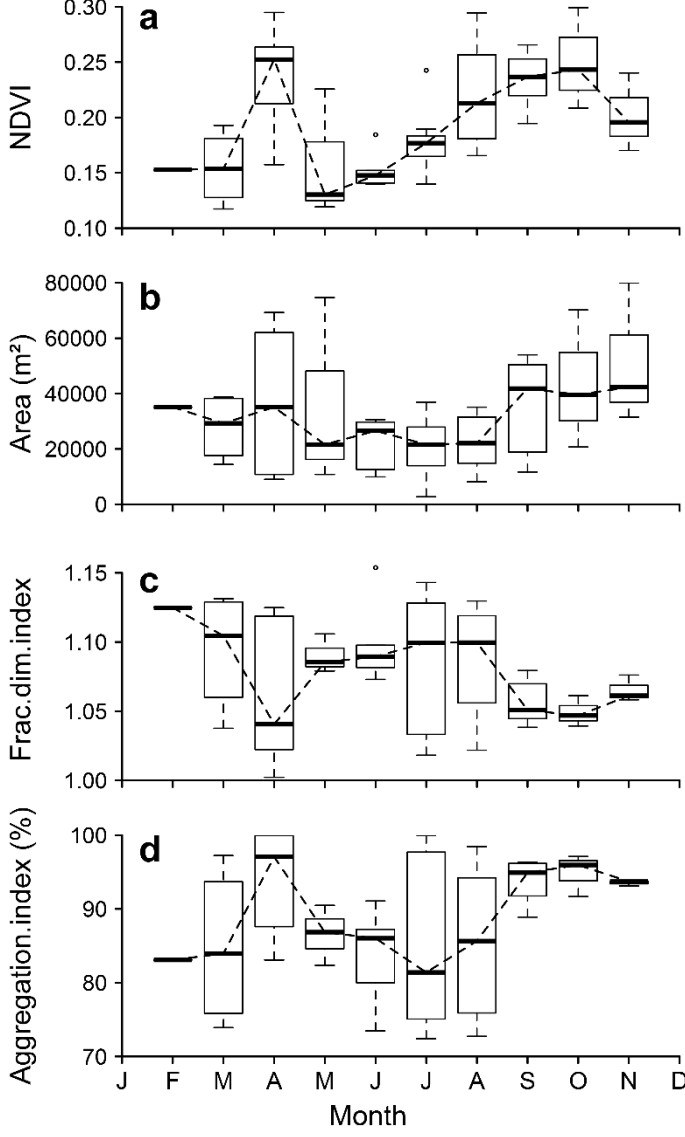

**Figure 6.** NDVI and spatial metrics of the control reef NDVI patch during 1985-2015: **(a)** NDVI monthly variation in the

control reef distance buffer, **(b)** patch area, **(c)** patch fractal dimension index and **(d)** patch aggregation index. Horizontal lines

denote the median value, boxes represent first and third quartiles, and whiskers represent the last value within 1.5 times the

interquartile distance.



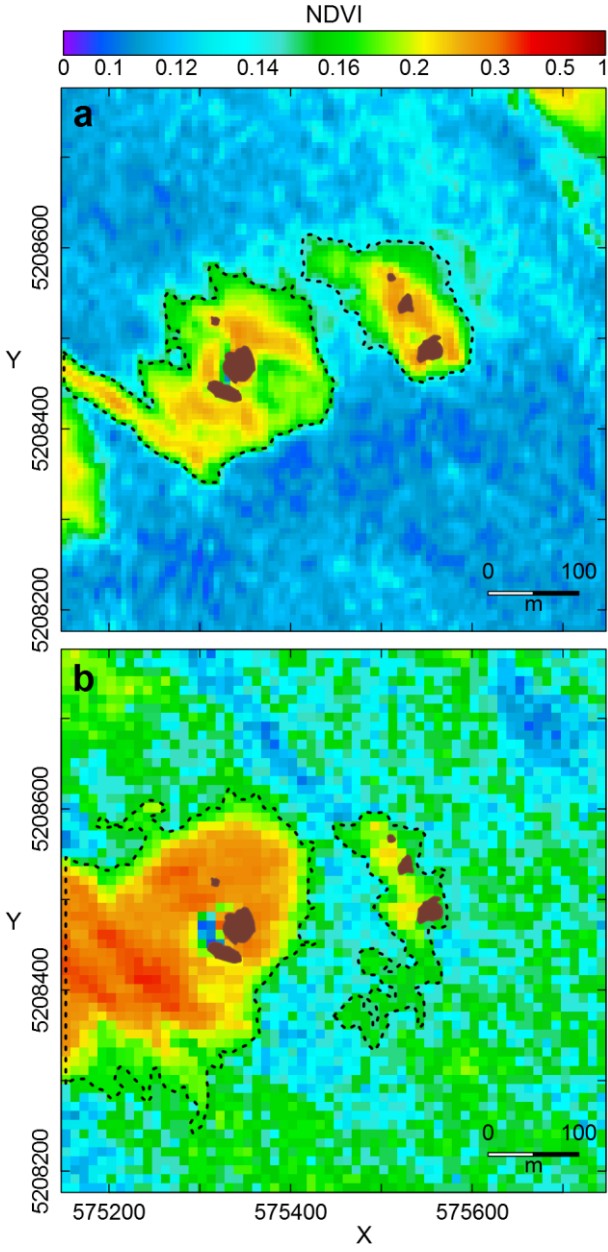

**Figure 7.** MPB patches (dashed lines) around the control reef and the impacted reef (both represented in brown) showing the temporal evolution of their spatial structure before and after the BACI experiment: **(a)** 11 months before the BACI experiment (SPOT 6 image acquired on 20th August 2013), **(b)** 3 months after the experiment (SPOT 5 image acquired on 9th October 2014). Images are displayed at their original spatial resolution (respectively 6 m and 10 m). Data were downscaled to 30 m for the analysis.



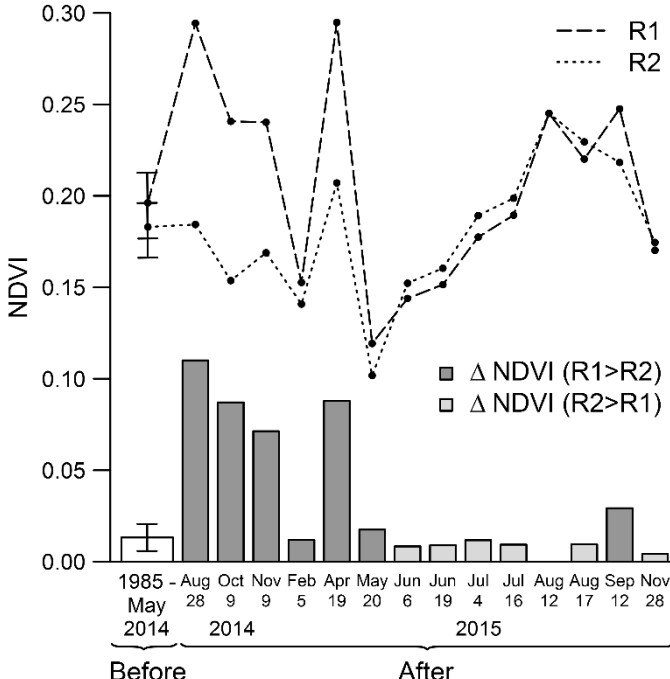

**Figure 8.** R1 control reef (dashed line), R2 impacted reef (dotted line) mean NDVI, and R1 and R2 ΔNDVI (grey bars). Mean NDVI were calculated for each image of the time series based upon the distance buffer derived from the spatial metrics. R1 and R2 first points corresponds to their respective mean NDVI before the BACI experiment (mean ± 95% CI). The white bar corresponds to the mean difference observed between the patches before the BACI experiment (mean ± 95% CI). The other values represent the means and differences observed between the patches on each image acquired after the BACI experiment.