# Peer review of "Satellite remote sensing reveals a positive impact of living oyster reefs on microalgal biofilm development"

_Biogeosciences, 2017_

## Referee Comment (RC1) · Anonymous Referee #1 · 10 Oct 2017

The work is a good contribution to the use of RS for mapping MPB mats, and useful in determining the zone of influence of wild oyster reefs on enhanced primary production

- the authors should mention, either from the literature, or from their own studies, how the biomass of the benthic diatoms (MPB) is related to the spectroradiometric ground truthing data. Can the biomass also be related to oyster biomass in some way? For example 1 kg of oysters = 1 kg of MPB

- the authors should estimate the area MPB of the impacted reef before and after, and if possible, relate that to the loss in algal biomass.

- the authors should check the MS for run-on sentences to make the MS easier to read.

- the authors might consider mentioning that the hydrodynamics (erosional currents and waves) in the vicinity of the oyster reef effect the net biodeposition and availability of nutrients to the MPB so the distribution might also be related to other factors besides just the presence/absence of live oysters

---

## Referee Comment (RC2) · Anonymous Referee #2 · 11 Oct 2017

The study by Echappe et al. is an interesting application of remote sensing. My research covers bivalve ecology, so I don't have expertise in the methodology used to compile and analyze the time series of satellite images. However, I found the explanation of the approach and its utility to be helpful in this manuscript. Monitoring variation in the abundance and distribution of benthic diatom assemblages (MPB) is a great application of the technology. The authors are up-front about the inherent limitations to the approach, but clearly show how it provides a hands-free platform for documenting long-term trends in the abundance and distribution of MPB; the extension of this technology for additional studies in ecology is truly exciting.

[Figure]

On the other hand, as an ecologist, there are aspects of the study by Echappe et al that are a bit concerning. Oysters are considered to be ecosystem engineers because of the role they play in benthic-pelagic coupling and the deposition of organic matter exported from an oyster reef has significant impact on the surrounding benthos. Thus, it is not surprising to see the halo effect of oyster reefs on MPB abundance and productivity. However, the BACI experiment designed to test the effect of reef structure versus oyster activity on MPB is unreplicated and the study did not include any sampling of the benthos to "ground-truth" the satellite imagery. Granted, large scale ecological experiments are not easy to replicate and I recognize that the BACI experiment described in this experiment falls into this scale of effort. Such large scale experiments are important even without replication but in discussing the results the authors need to recognize the limitations of their unreplicated designs. For example, on line 285 they assert, "our data thus strongly suggest that the presence of live oyster reefs promotes MPB biomass development and affects MPB spatial distribution around the reef". To me, the authors are overreaching with such assertions and the abstract does not include any mention of the potential confounding effects of sediment disturbance. Instead, the authors comment that the BACI results "confirmed" their conclusions from the longer time series analysis of satellite imagery. The lack of replication means that the authors cannot generalize their findings in this way.

Certainly there is an association with the reef and MPB dynamics as seen in the satellite images. On line 329 they discuss the potential confounding effects disturbance to the sediment community by burning the oyster reef, but this is mostly hand-waving and not a very effective discussion of the confounding effects of the disturbance created by burning the reef. Sadly, there was also an opportunity lost in the BACI project. Searching the literature for information on how disturbance affects MPB communities, most of the focus has been on sedimentation and resuspension events, or toxic pollutant (e.g., oil spill) impacts on MPB biomass. It would have been highly informative to periodically sample the MPB, examine whether any shifts in abundance or dominance had occurred (as often happens in many ecological communities) post disturbance

and how such changes affected overall biomass development. I'm not an expert on MPB communities, but studies by Blanchard et al. (2000; Continental Shelf Research 20:1243-1256) and other indicate that MPB activity helps stabilize mudflats which, in turn, helps promote MPB community development. Ground-level sampling could have helped to ascertain whether burning impact stability of the community and thus, how much of a role community diversity and stability impacted recovery time.

This manuscript should go forward, but there needs to be more recognition of the weakness on the ecology side of the study.

---

## Author Comment (AC1) · 22 Nov 2017

**Response to the referees**

We would like to thank both reviewers for their time and their helpful comments and suggestions. Please find below a detailed response to each reviewer. The reviewers' comments are highlighted in bold and our responses are in blue, normal text.

**Anonymous Referee #1**

**RC1: The work is a good contribution to the use of RS for mapping MPB mats, and useful in determining the zone of influence of wild oyster reefs on enhanced primary production.**

We appreciate your valuation of our work.

**RC1: the authors should mention, either from the literature, or from their own studies, how the biomass of the benthic diatoms (MPB) is related to the spectroradiometric ground truthing data. Can the biomass also be related to oyster biomass in some way? For example 1 kg of oysters = 1 kg of MPB**

*In situ* MPB biomass measurements allowing to groundtruth our NDVI data with actual MPB biomass values (e.g. in g.chla/m²) could not be performed in the framework of our specific study. However, NDVI is a very commonly used proxy for MPB biomass, and different authors already established NDVI – MPB biomass relationships using either satellite data (Brito et al., 2013) or hyperspectral reflectance data (Méléder et al., 2003).

Regarding the relationship between MPB and oyster biomass, we currently do not have any data allowing us to relate MPB biomass to oyster biomass and are not aware of studies who have actually attempted to do this.

References cited:

Brito, A.C., Benyoucef, I., Jesus, B., Brotas, V., Gernez, P., Mendes, C.R., Launeau, P., Dias, M.P., Barillé, L., 2013. Seasonality of microphytobenthos revealed by remote-sensing in a South European estuary. Continental Shelf Research 66, 83–91. doi:10.1016/j.csr.2013.07.004

Méléder, V., Barillé, L., Launeau, P., Carrère, V., Rincé, Y., 2003. Spectrometric constraint in analysis of benthic diatom biomass using monospecific cultures. Remote Sensing of Environment 88, 386–400. doi:10.1016/j.rse.2003.08.009

**RC1: the authors should estimate the area MPB of the impacted reef before and after, and if possible, relate that to the loss in algal biomass.**

This would undeniably be a simple, direct information to present, yet we do not think it would show any result relevant to the contents of the manuscript. Information given by the patch metrics highlighted the fact

that the extension of the MPB area is not related to MPB biomass in a straightforward way. Indeed, highly concentrated patches tend to be more compact while a loss in biomass can go together with a loss in aggregation, leading to a wider area with lower values. This is one of the reasons why we focused on concentrations instead of areas, and in relative differences between the impacted MPB area and the control one.

**RC1: the authors should check the MS for run-on sentences to make the MS easier to read.**

This suggestion is quite vague however welcome. We tried to take it into account by modifying the following sentences in the manuscript:

L.46: "In Colden et al. (2016), experimental oyster reefs consisting of empty shells only were shown to modify the local hydrodynamic conditions.  They also promote the trapping of fine particles, providing conditions more conducive to benthic microalgal development."

L.58: "Few RS studies to date however have described MPB dynamics at both high spatial and temporal resolution. This can be explained by the reduced availability of high spatial resolution satellite time series together with the constraints related to the acquisition of RS data during low tide only."

L.261: "NDVI then gradually recovered to its usual level in the following months.  Even higher NDVI values than the pre-experiment average  about one year later, from July to September 2015, despite the absence of a clearly identifiable patch structure around R2."

L.268: "The difference ratio is especially striking during the first months after the killing, considering that NDVI values around R1 were exceptionally high in 2014 compared to the seasonal average One would therefore expect R2 NDVI values to also stand above average."

L.337: "The use of a RS historical time series (29 years of data before the experiment and more than one year of data after it) allowed for a simultaneous and synoptic collection of data for both the control and impact sites It also enabled the differentiation of seasonal and interannual variability from the impact of the experiment."

L.380: "Although it does not allow to identify the cause of the patterns observed, it provided a sufficiently explicit spatial tool able to describe MPB structure at mesoscale.  The length of the time series also made it possible to quantify the size of the detectable reef footprint (Giles et al., 2009) through the NDVI in our study."

**RC1: the authors might consider mentioning that the hydrodynamics (erosional currents and waves) in the vicinity of the oyster reef effect the net biodeposition and availability of nutrients to the MPB**

**so the distribution might also be related to other factors besides just the presence/absence of live oysters.**

This is a very relevant suggestion as hydrodynamics have been shown to play a role in making nutrients available to the MPB, and hydrodynamics are affected in the vicinity of oyster reefs. We however draw your attention to the fact that we do mention the effects of hydrodynamics in relation to oyster reefs in the following paragraph, as part of the discussion section:

L.317-324: "Oyster reefs are also known to have indirect effects such as modifying the structure of the surrounding sediment and the ambient hydrodynamic conditions, facilitating MPB establishment (Colden et al., 2016). In a recent paper on the impact of mussel beds on MPB biomass development, Engel et al. (2017) attributed the positive effect of such beds on MPB biomass to a combination of reduced hydrodynamic stress and increased nutrient availability (and also to potential changes in the associated invertebrate community). In contrast with the observations of Colden et al. (2016) and Engel et al. (2017), our BACI experiment, in which the oysters were killed while the physical structure of the reef itself was not modified, now allows distinguishing the pure physical (hydrodynamic) effect from the biological (nutrient enrichment) effect of the oyster reefs, suggesting that the latter process is more important in our study area."

As mentioned at the end of the above paragraph, the specific approach used in our experiment (which makes it possible to distinguish the hydrodynamics effects from the biological effects because the physical structure of the reef was kept intact. Therefore, while acknowledging the possibility of other factors influencing MPB spatial distribution, this allowed us to focus specifically on the impact of the biological effect, i.e. live oysters.

**Anonymous Referee #2**

**RC2: The study by Echappe et al. is an interesting application of remote sensing. My research covers bivalve ecology, so I don't have expertise in the methodology used to compile and analyze the time series of satellite images. However, I found the explanation of the approach and its utility to be helpful in this manuscript. Monitoring variation in the abundance and distribution of benthic diatom assemblages (MPB) is a great application of the technology. The authors are up-front about the inherent limitations to the approach, but clearly show how it provides a hands-free platform for documenting long-term trends in the abundance and distribution of MPB; the extension of this technology for additional studies in ecology is truly exciting.**

We thank you for your thorough reading of our manuscript as well as for your appreciation and thoughtful comments.

**RC2: On the other hand, as an ecologist, there are aspects of the study by Echappe et al that are a bit concerning. Oysters are considered to be ecosystem engineers because of the role they play in benthic-pelagic coupling and the deposition of organic matter exported from an oyster reef has significant impact on the surrounding benthos. Thus, it is not surprising to see the halo effect of oyster reefs on MPB abundance and productivity. However, the BACI experiment designed to test the effect of reef structure versus oyster activity on MPB is unreplicated and the study did not include any sampling of the benthos to "ground-truth" the satellite imagery. Granted, large scale ecological experiments are not easy to replicate and I recognize that the BACI experiment described in this experiment falls into this scale of effort. Such large scale experiments are important even without replication but in discussing the results the authors need to recognize the limitations of their unreplicated designs. For example, on line 285 they assert, "our data thus strongly suggest that the presence of live oyster reefs promotes MPB biomass development and affects MPB spatial distribution around the reef". To me, the authors are overreaching with such assertions and the abstract does not include any mention of the potential confounding effects of sediment disturbance. Instead, the authors comment that the BACI results "confirmed" their conclusions from the longer time series analysis of satellite imagery. The lack of replication means that the authors cannot generalize their findings in this way.**

Our experiment could indeed not be replicated, partly because of logistical reasons (the distance between the oyster reefs and the shore made the field work in the mudflat very laborious), partly because of the size of the experiment (two reefs with more than 25 tons of oysters). We are aware of the lack of replication, and we decided to compensate this by the creation of a long-term, robust dataset. We are happy that you recognized that "Such large scale experiments are important even without replication". However, it is true that our assertions might seem too strong in that regard, therefore we modified the manuscript with more careful statements:

L.19: "This observation was then  supported by the BACI experiment"

L.283: "Killing of oysters in one reef  as part of a BACI experiment  highlighted the positive impact of live oysters on MPB biomass development and dynamics. Taken together, our data thus  suggest that the presence of live oyster reefs promotes MPB biomass development and affects MPB spatial distribution around the reef."

We also modified and reorganized the following paragraph to make the questions raised by the lack of replication clearer:

L.337-345: "The use of a RS historical time series (29 years of data before the experiment and more than one year of data after it) allowed for a simultaneous collection of data for both the control and impact sites, and for the differentiation of seasonal and interannual variability from the variability due to the impact of the experiment. The paired monitoring of the control and impacted reefs confirmed a low inter-site variability, while their proximity submitted them to similar environmental conditions (Stewart-Oaten et al., 1986). Unfortunately, the nature, scale and location (nearly one km from the shore) of the experiment made replication not possible. However, while we recognize the limitation of the unreplicated design, we believe that our experimental design is suitable to distinguish site-specific NDVI variations from the average mudflat trend (Hewitt et al., 2001), as (1) satellite data of the control site showed trends consistent with whole mudflat MPB temporal dynamics, and (2) they also revealed no response related to the experiment, underscoring their reliability as control values."

Concerning the MPB ground truthing, we refer to our answer to the second remark of RC1, who made a similar comment. We also emphasize the fact that we validated our satellite NDVI data by performing MPB ground truth radiometric measurements.

**RC2: Certainly there is an association with the reef and MPB dynamics as seen in the satellite images. On line 329 they discuss the potential confounding effects disturbance to the sediment community by burning the oyster reef, but this is mostly hand-waving and not a very effective discussion of the confounding effects of the disturbance created by burning the reef. Sadly, there was also an opportunity lost in the BACI project. Searching the literature for information on how disturbance affects MPB communities, most of the focus has been on sedimentation and resususpension events, or toxic pollutant (e.g., oil spill) impacts on MPB biomass. It would have been highly informative to periodically sample the MPB, examine whether any shifts in abundance or dominance had occurred (as often happens in many ecological communities) post disturbance and how such changes affected overall biomass development. I'm not an expert on MPB communities, but studies by Blanchard et al. (2000; Continental Shelf Research 20:1243-1256) and other indicate that MPB activity helps stabilize mudflats which, in turn, helps promote MPB community development. Ground-level sampling could have helped to ascertain whether burning impact stability of the community and thus, how much of a role community diversity and stability impacted recovery time.**

We fully agree with this remark but unfortunately such samples were not taken as the sampling site is very inaccessible (which is also the reason why we investigated the use of RS for monitoring the MPB dynamics). Data on community structure would undoubtedly yield more insight into the nature of the observed changes related to seasonality and the experimental treatment.

**RC2: This manuscript should go forward, but there needs to be more recognition of the weakness on the ecology side of the study.**

We thank you for your recommendation and truly hope that we have sufficiently addressed your concerns about the potential ecological limitations of our work.